# Effect of Free Long-Chain Fatty Acids on Anagen Induction: Metabolic or Inflammatory Aspect?

**DOI:** 10.3390/ijms26062567

**Published:** 2025-03-13

**Authors:** Xiaowen Pan, Khava S. Vishnyakova, Elina S. Chermnykh, Maxim V. Jasko, Alexander D. Zhuravlev, Svetlana S. Verkhova, Yegor S. Chegodaev, Mikhail A. Popov, Nikita G. Nikiforov, Yegor E. Yegorov

**Affiliations:** 1Engelhardt Institute of Molecular Biology, Russian Academy of Sciences, 32 Vavilov Street, Moscow 119991, Russia; panxiaowen1995@mail.ru (X.P.); khava58@mail.ru (K.S.V.); 2003_maxim@mail.ru (M.V.J.); egozavr-ch@mail.ru (Y.S.C.); nikiforov.mipt@gmail.com (N.G.N.); 2Koltzov Institute of Developmental Biology, Russian Academy of Sciences, 26 Vavilov Street, Moscow 119334, Russia; elinachermnykh@mail.ru; 3Laboratory of Angiopathology, Institute of General Pathology and Pathophysiology, 8 Baltiyskaya Street, Moscow 125315, Russia; zhuravel17@yandex.ru (A.D.Z.); verxova.svetlana@gmail.com (S.S.V.); popovcardio88@mail.ru (M.A.P.); 4Laboratory of Cellular and Molecular Pathology of Cardiovascular System, Federal State Budgetary Scientific Institution “Petrovsky National Research Centre of Surgery”, 3 Tsyurupa Street, Moscow 117418, Russia; 5Department of Cardiac Surgery, Moscow Regional Research and Clinical Institute (MONIKI), 61/2, Shchepkin Street, Moscow 129110, Russia; 6Center for Precision Genome Editing and Genetic Technologies for Biomedicine, Institute of Gene Biology, Russian Academy of Sciences, 34/5 Vavilov Street, Moscow 119334, Russia

**Keywords:** hair growth, free fatty acids, lactate, glycolysis, inflammation, metabolic reprogramming

## Abstract

Hair growth is a highly complex process regulated at multiple levels, including molecular pathways, stem cell behavior, metabolic processes, and immune responses. The hair follicle exhibits metabolic compartmentalization, with some cells relying on glycolysis and others on oxidative phosphorylation. Interestingly, in mice, the onset of the anagen phase can be stimulated by locally suppressing oxidative phosphorylation in the skin. This study showed that topical application of palmitate or oleate accelerated the onset of anagen in mice, while lactate, the end product of glycolysis, delayed it. We also investigated the effects of fatty acids on cytokine production in various human cell cultures. Fatty acids did not induce a cytokine response in fibroblasts or keratinocytes but significantly affected monocytes. Specifically, palmitic acid induced the production of TNF-α, IL-8, and CCL2. Oleic acid, however, elicited almost no response. By comparing the “metabolic” and “inflammatory” hypotheses of anagen stimulation, the results of our study suggest that metabolic regulation holds significant promise for influencing hair growth.

## 1. Introduction

Hair growth is an extraordinarily complex process involving a variety of factors, including cellular aging processes, regulation of various stem cells and their niches, hormonal regulation, and others [1,2].

The regulation of hair growth occurs locally (within the hair follicle), as well as within the pilosebaceous unit, at the level of the skin area, and at the level of the whole organism. This makes it very difficult to study and select experimental models [3].

A clear link between the immune response and hair regeneration has been traced in [4]. The process of hair growth is an example of true regeneration, and regeneration is always directly related to inflammatory processes and the activity of the immune system [5,6,7]. The effects on hair growth of lymphocytes, macrophages, adipose tissue, and fibroblasts have been described [8,9,10,11,12,13]. Although the immune system is involved in the regulation of hair growth, the hair follicle itself is an immune-privileged site. This is achieved in various ways: by the restriction of immune cell recruitment, reduction of antigen presentation by the decreased expression of MHC I, MHC II, and β2-microglobulin, and active immunosuppression by the production of IL-10, TGFb1, TGFb2, and MHC I immunoregulatory molecules HLA-G and HLA-E [14].

Hair growth occurs in repetitive cycles. There is a growth stage, anagen, followed by a degeneration stage, catagen, passing into telogen—a resting stage [15].

One of the characteristics of hair growth is that this process is unusually active and thus highly metabolically dependent. Hair growth begins with preparatory processes of energy and material storage. To begin anagen, the hair follicle descends into the thickness of specially formed adipose tissue; its blood supply develops. The hair cycle in humans can be several years; in mice, it is much shorter (about a month). In rodents, in particular mice, unlike humans, synchronization of follicle growth in different skin areas is observed, which is accompanied by cyclic changes in skin thickness due to the development and degradation of adipose tissue [16].

Obviously, in the case of the hair follicle, effects on cell metabolism should strongly affect hair growth. Metabolism largely determines the properties of cells [17].

We have long studied the effects of free long-chain fatty acids on hair growth in mice. It was shown that in aging mice, application of fatty acids resulted in overgrowth of bald spots. With prolonged application to the skin in mice, the skin thickens due to subcutis and the size of hair follicles increases [18,19].

Our previous works were conducted on adult and old mice, where hair growth waves are already poorly expressed. We decided to repeat the experiments using other mouse models. To facilitate the analysis, we decided to perform the classical work on young mice (between the first and second growth waves) as well as on mice with a mutation resulting in extremely short hair. In this case, hair growth waves (and thus their stimulation) can be observed without resorting to depilation.

Since fatty acids are applied to the skin, which contains different cell types, we do not know the possible acceptors of the effect. Therefore, we decided to test the effect of fatty acids on keratinocytes, fibroblasts, and monocytes. The metabolic effects of fatty acids on different cells have been studied previously [20]; in the present work, we tested the possibility of pro- or anti-inflammatory effects of fatty acids on different cells.

## 2. Results

### 2.1. Experiments with we/we wal/wal Mice

The we/we wal/wal mice were chosen because of their hair characteristics. The spontaneous homozygous we (wellhaarig) mutation, characterized by a wavy coat and curly whiskers, results from a defect in epidermal transglutaminase 3 (Tgm3) [21]. The Tgm3 mutation alters the crosslinking of proteins in the hair cortex and may therefore affect the hair phenotype [21]. The molecular nature of the genetic defect in wal (waved alopecia) is still unknown; however, the coordinates of the chromosomal locus containing the wal gene have been determined on mouse chromosome 14 [22]. Genome-wide sequencing of wal/wal mice found only an abnormality outside the wal zone, a mutation in the Slc9a9 gene, which is unlikely to be associated with the wal phenotype [23]. Phenotypically, the wal mutation in a homozygous recessive state is manifested by a wavy coat. Over time, partial baldness develops, leading to a thinning of the coat in mice. More severe abnormalities in hair follicle morphogenesis were found in we/we wal/wal double mutants; thus, Tgm3 has a modifying effect on the wal gene, increasing alopecia in the double mutants. The mutations that occurred in these mice did not allow the hair to grow long [24,25,26], so the skin was always visible through the hair, allowing us to track hair follicles in the anagen stage. It is known that at the onset of anagen in black mice the skin turns black, but in normal mice, we do not see this because the hair coat completely covers the skin. Thus, we/we wal/wal mice are like permanently shaved black mice.

Figure 1 shows the results of the fatty acids mixture exposure of a group of not old (approximately 6 months old) mice.

In the experiment, we see a more contrasting pattern on the skin with darker patches. This is evidence of anagen stimulation as a result of our treatment. We repeated these experiments twice on four older mice. They were approximately 1 year old at the beginning of the experiments. Twice we obtained the same result: after about 17 days from the beginning of the experiments, we observed an increase in skin contrast with the appearance of dark areas corresponding to anagen (Figure 2).

Thus, the action of the fatty acid mixture stimulates anagen in mice of different ages.

### 2.2. Experiments with Young C57Bl/6 Mice

Due to the limitations associated with poor reproduction of we/we wal/wal mice, we continued with the conventional model of the onset of anagen in C57BL/6 mice. For this purpose, at 45–48 days of age, we depilated the hair on the back of the mice and treated the skin of the mice every other day with different fatty acids or a mixture. Palmitic and oleic acids were found to significantly stimulate the onset of anagen (Figure 3). The mean ages at which the dorsal skin of mice dabbed with palmitic and oleic acids began to gray were 52 and 53 days, while in the control group this was 56 days (Figure 3). These fatty acids apparently stimulate the initiation of the hair growth cycle in mice. Other fatty acids (linolenic acid and a mixture of fatty acids) also showed a trend toward anagen stimulation, but this trend was not significant.

### 2.3. Lactate Experiments

It is known that the activity of lactate dehydrogenase (Ldha) strongly influences the activation of hair growth. Studies have shown that selective deletion of the *Ldha* gene in follicle cells blocks follicles in the telogen, while enhancement of glycolysis by blocking the pyruvate transporter (Mpc 1) or pharmacological induction of Ldha causes the transition from telogen to anagen [27].

In order to influence the activity of lactate dehydrogenase in skin cells, we decided to conduct experiments with lactate. For this purpose, we treated the skin of mice with a 13% solution every other day, as in the case of fatty acids.

As a result, it turned out that such treatment inhibited the beginning of anagen (Figure 4). At the same time, we did not observe the toxic effect of lactate: some mice, despite the treatment, developed black coloration and hair growth (one of the mice in Figure 4a, indicated by the arrow).

### 2.4. Effect of Fatty Acids on the Production of Inflammatory Factors by Cells in Culture

Hair growth activation is closely related to inflammation processes [5,6,7]. We decided to test whether fatty acids cause induction of inflammatory factors in cells of various differentiation lines: fibroblasts, keratinocytes, and monocytes.

Experiments showed that fibroblasts and keratinocytes had little or no response to the addition of different fatty acids. In monocytes, we observed pronounced responses (Figure 5). Palmitic acid significantly induced the synthesis of TNF-a, IL-8, and CCL2. In the case of IL-6, we observed a tendency toward activation; in the case of IL-1b, there was no activation. The mixture of fatty acids had no significant effect.

## 3. Discussion

The data obtained showed that fatty acid application induces the onset of anagen waves in we/we wal/wal mice as well as C57Bl/6 mice. This is consistent with previous findings of induction of overgrowing of bald spots in aging mice by fatty acid application [18,19]. A total of three different models showed induction of hair growth by fatty acid treatment: the model of induction of the second growth wave in seven-week-old mice (the most common model), multiple induction of growth waves in mature and aging mice, and induction of overgrowing of bald patches in aging mice.

Anagen stimulation was obtained using both palmitate and oleate, two fatty acids with very different properties. Palmitate is known to be highly toxic to cells [28,29], whereas oleate can even rescue cells from the toxic effects of palmitate [30,31,32].

As we have shown previously, palmitate and oleate share the property of enhancing glycolysis in various cells [20]. Induction of glycolysis by palmitate has also been recently described [33].

A number of works indicate that activation of hair follicle stem cells occurs with an obligatory increase in glycolysis and lactate dehydrogenase activity [27,34]. Moreover, suppression of energy generation by mitochondria, which should inevitably increase glycolysis, also causes induction of anagen [35,36].

Although hair follicles contain a sufficient number of functioning mitochondria, up to 90% of glucose is metabolized to lactate despite the presence of oxygen [37]. However, quite different metabolism exists in different parts of the follicle [38]. Hair follicles have a functioning internal Cori cycle and glycogen stores that are consumed during growth [39].

If the assumption of the role of lactate dehydrogenase (glycolysis) is correct, then suppression of glycolysis should inhibit the onset of anagen. This was the case (Figure 4). It is known that lactate suppresses glycolysis [40,41,42] and, moreover, stimulates the work of mitochondria (electron transfer, respiration, ATP synthesis) [42].

It can be suggested that the inhibitory effect of lactate is explained simply by its toxic effect, but in the experiments with lactate, we observed a part of mice in which hair grew perfectly even during lactate treatment (Figure 4, arrow). Most likely, in such mice, the decision to start the wave of hair growth occurred even before the beginning of the experiment.

The weak point of the growth wave model was the significant asynchrony of the mice. Based on many publications [15,27,35,36], we assumed that all mice aged 45–48 days would be synchronized between the first and second growth waves and follicles would be in telogen. This turned out not to be the case. We did not include approximately 37% of the mice in the experiments after depilation revealed areas of dark (anagen) skin. It is possible that some of the mice included in the experiments had a second growth wave already induced but not yet manifested. Such a feature explains the large variation in data in the experiments.

The hair cycle involves elements of inflammation. We tested the possible induction of inflammatory intermediates by fatty acids. The results showed that, indeed, palmitate causes the induction of a number of cytokines, but this ability was not possessed by oleate. Given that both palmitate and oleate stimulate the onset of second anagen, it is reasonable to assume that the mechanism of this action is not related to the induction of inflammation.

Although oxidative metabolism is more efficient in terms of energy generation, extremely intense hair growth requires an array of metabolites that are produced during glycolysis. It is hypothesized that metabolism itself may play a role as a switch between quiescence and proliferation. Metabolic reprogramming may occur [27,43,44,45,46].

The main limitation of this work is that the results were obtained in a mouse model of hair growth and may be of limited applicability to stimulation of hair growth in humans.

## 4. Materials and Methods

### 4.1. Mice

The study was approved by the local Bioethics Committee of the Koltzov Institute of Developmental Biology (Protocol No. 56, 4 February 2022). All experiments were performed in compliance with the guidelines for the care and use of laboratory animals established at the Institute of Developmental Biology. We used wild type mouse strain C57Bl/6 and double homozygote mutant mice we/we wal/wal. The mutant mice were generated by the crossing of we/we wal/wal and C57Bl/6 background mouse strains and kindly provided by Koniukhov and Nesterova [24,25,26].

Twenty-four young and four old mice we/we wal/wal and fifty C57Bl/6 mice participated in the experiments.

### 4.2. Experiments on C57BL/6 Mice

Hair was depilated with depilatory cream (Veet, for sensitive skin, India) on the backs of mice from the shoulder blades to the middle of the tail (approximately 4cm^2^), at 45–48 days of age. Only mice in the telogen stage (pink skin) were taken for further work. Mice with gray skin areas made up about 37% of the total. They were not used in further work. Mice were divided into experimental and control groups and were treated every second day (3 times a week) throughout the experiments. The solutions of either fatty acid or unrefined virgin olive oil (control), or lactate (100 μL) were applied on the back and then spread. Photographs were taken to illustrate the change in coloration.

We determined the day on which a mouse began to gray by comparing the changes in its skin color over three days: the day before graying began, the day the skin began to turn grey, and the day after graying (Figure 6). Once the mice began to turn grey, predictable changes occurred each day.

### 4.3. Fatty Acids

The different fatty acids used were: palmitic acid, C16:0 saturated, (“Sigma-Aldrich”, P0500, Burlington, MA, USA); oleic acid, C18:1, cis, omega 9 (“Sigma-Aldrich” O1630, Burlington, MA, USA); linolenic acid, C18:3, omega 3, cis, cis, cis 9,12,15 (“Sigma-Aldrich” L2376, Burlington, MA, USA); and linoleic acid, C18:2 omega 6, cis, cis 9,12 (“Sigma-Aldrich” L8134, Burlington, MA, USA).

By combining different fatty acids, we obtained a mixture distantly resembling the mixture of fatty acids in human blood plasma [47]. This mixture consists of 65% oleic, 22% linoleic, 10% palmitic, and 3% linolenic acid. It was used in experiments on mice.

In experiments on mice, we used 10% solutions of fatty acids in unrefined virgin olive oil.

Fatty acids are very poorly soluble in water, so we prepared complexes of fatty acids with bovine serum albumin to work with cells: we mixed a hot alcoholic solution of fatty acids (60 °C) with a warm (37 °C) solution of bovine serum albumin (7–10%). The pH of the solution was then adjusted to 7.2–7.4 using 10% sodium hydroxide. The solution was filtered through a 0.2 µm pore diameter filter, divided into aliquots, and stored at −20 °C.

A mixture of lactic acid and its salt (hereinafter lactate) was prepared as follows: a concentrated 80% solution of lactate (Terra Aromatica, Moscow, Russia) was diluted with water and the pH was adjusted to 5.5–6 with 10 M KOH (a slightly acidic pH is considered normal for the skin surface). Next, glycerol was added. The final solution contained 13% lactate and 72% glycerol in water.

### 4.4. Cells

The following cell cultures were used: 977 hTERT—a strain of human embryonic fibroblasts expressing the protein component of telomerase [20] and the HaCat line of immortalized human keratinocytes. Donor human monocytes were also used. Monocytes were prepared as described in [48]. Blood samples were collected from 6 healthy volunteers aged 20–30 years. All volunteers provided informed consent to participate in the study, which was approved by the Ethics Committee of the Moscow Regional Research and Clinical Institute named after M.F. Vladimirsky (Ethics Protocol No. 12/2021). All volunteers had no chronic diseases and did not take any medications, alcohol, or narcotic substances.

Primary monocytes were isolated from whole blood using Ficoll density gradient centrifugation (PanEco, Moscow, Russia) and subsequent sorting of CD14+ cells with CD14+ MicroBeads (Miltenyi Biotec, Bergisch Gladbach, Germany). Monocytes were cultured in 24-well plates (Corning Costar TC-Treated Multiple Well Plates, Sigma-Aldrich, USA) at a density of 1 × 10^6^ cells in 1 mL of X-VIVO 10 serum-free culture medium (Lonza, Basel, Switzerland).

Cells (except for human monocytes) were cultured at 37 °C in an atmosphere of 5% carbon dioxide in DMEM medium (PanEco, Moscow, Russia) containing 4.5 g/L glucose and supplemented with glutamine, gentamicin (40 μg/mL), and 10% fetal bovine serum (Biolot, St. Petersburg, Russia). Experiments to study the effect of fatty acids were performed in complete medium with serum. Samples of cell culture supernatants were stored at −80 °C and thawed before immunoenzyme analysis.

### 4.5. Evaluation of Cytokine Secretion

The levels of cytokines TNF-a, IL-1beta, IL-6, and IL-10, and chemokines IL-8 and CCL2 were measured in cell culture supernatants using enzyme-linked immunosorbent assay (ELISA) according to the manufacturer’s instructions (R&D Systems, Minnneapolis, MN, USA).

Fatty acids as conjugates with BSA at concentrations of 100 μM (palmitic and oleic) were added for 24 h in complete medium with serum.

An immunoassay was performed in a 96-well plate (SPL, Cat. No. 32396) using DuoSet ELISA Development System reagents. Antibodies to TNF-α (R&D Systems, DY210), IL-1β (R&D Systems, DY201), IL-6 (R&D Systems, DY206), IL-8 (R&D Systems, DY208), IL-10 (R&D Systems, DY217B), and CCL2 (R&D Systems, DY279) were added to the wells of the plate and left overnight for adsorption. The next day, ELISA was continued according to the assay protocol. The optical densities of calibrators and test samples were measured on a plate reader (ClarioStar Plus, BMG Labtech, Ortenberg, Germany) at wavelengths of 450 and 570 nm. Calibration graphs were plotted on the basis of optical density values, which were used to calculate the concentrations of cytokines under study in pg/mL.

### 4.6. Statistical Analysis

Statistical analyses were performed using IBM SPSS Statistics 26 software for independent samples *t* tests (for lactate experiments) and a one-way ANOVA test for all other cases. We consider the differences to be statistically significant at *p*  <  0.05.

## Figures and Tables

**Figure 1 ijms-26-02567-f001:**
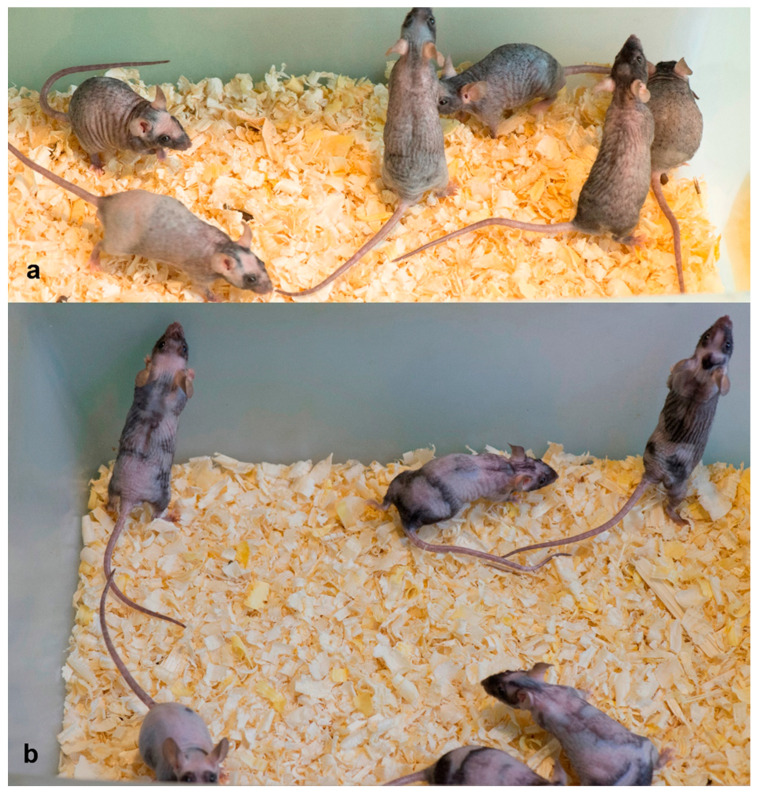
Effect of fatty acid mixture on the appearance of we/we wal/wal mice. (**a**)—control, (**b**)—mice after 17 days from the beginning of treatment. It can be seen that the treatment causes an increase in the contrast of skin coloration.

**Figure 2 ijms-26-02567-f002:**
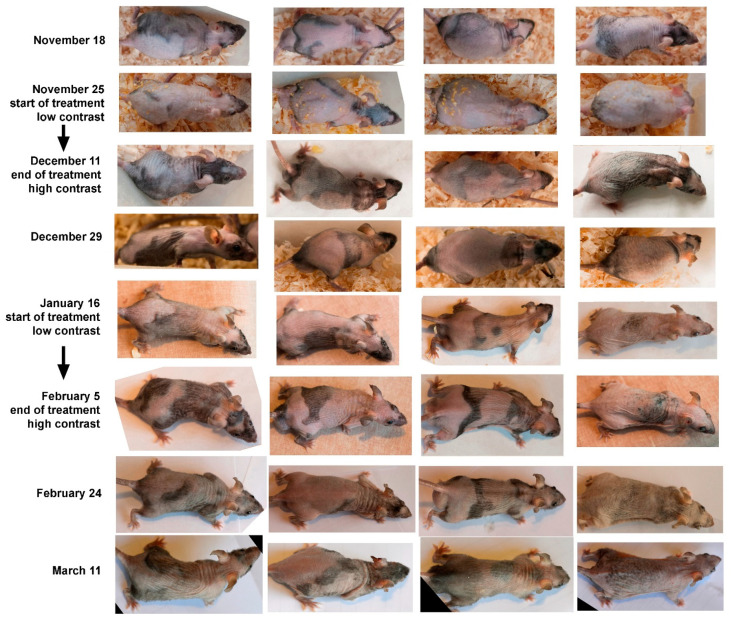
Effects of fatty acid mixture on the appearance of aged we/we wal/wal mice. Individual mice are represented vertically. Treatments were performed twice (labeled on the left). Both treatments resulted in enhancement of the dark components of the skin pattern.

**Figure 3 ijms-26-02567-f003:**
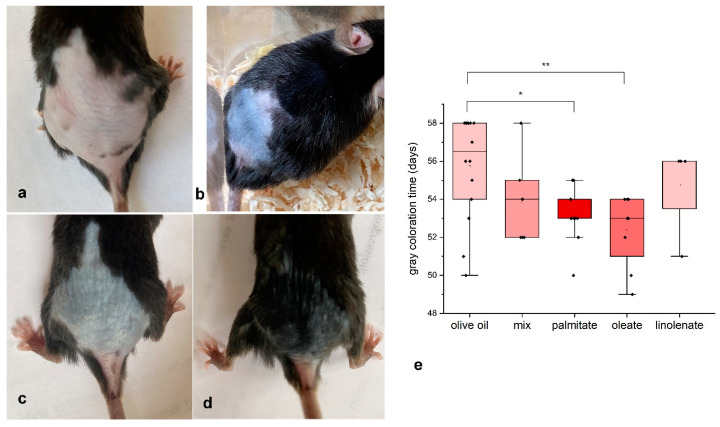
Effect of fatty acid treatment on the onset of anagen in C57Bl/6 mice. (**a**)—control, 57 days old; (**b**)—palmitic acid, 54 days old; (**c**)—oleic acid, 54 days old; (**d**)—same mouse as in 3c at 57 days old; (**e**)—graph of the appearance of gray skin coloration when treated with different fatty acids. * *p* < 0.05, ** *p* < 0.01.

**Figure 4 ijms-26-02567-f004:**
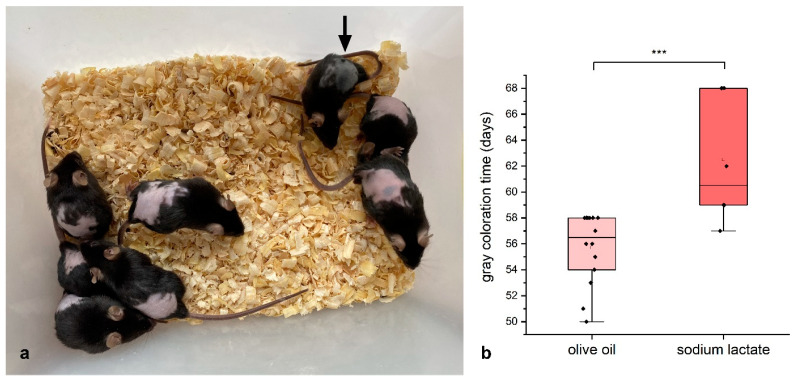
The effect of lactate on the onset of the anagen in C57Bl/6 mice. (**a**)—mice treated with lactate that reached the age of 60 days. The arrow shows a mouse in anagen. (**b**)—graph of the appearance of gray coloration of the skin. *** *p* < 0.005.

**Figure 5 ijms-26-02567-f005:**
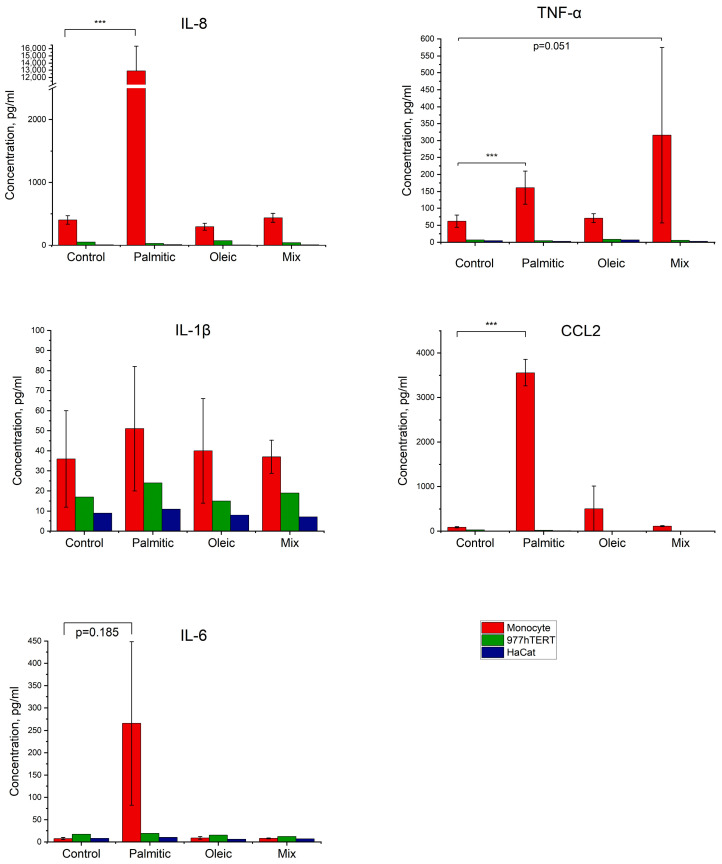
Induction of inflammatory factors in various human cells by fatty acids. Medians and standard deviations are shown. *** *p* < 0.005.

**Figure 6 ijms-26-02567-f006:**
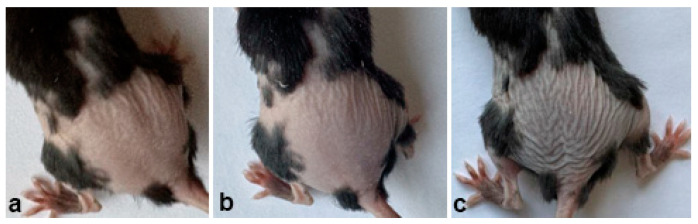
The process of skin color change during induction of anagen. (**a**)—is the day before the mouse turned gray, (**b**)—is the day when it started to turn gray, (**c**)—is the day after it started to turn gray. It can be seen that there is a clear difference.

## Data Availability

The data generated during the current study are available from the corresponding author on reasonable request.

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
