# Peer review of "Effect of Free Long-Chain Fatty Acids on Anagen Induction: Metabolic or Inflammatory Aspect?"

_ijms, 2025, doi:10.3390/ijms26062567_

Round 1
Reviewer 1 Report
Comments and Suggestions for Authors
Please, see document with the somment and suggestions attached.

Reviewer 2 Report
Comments and Suggestions for Authors
The paper presents an interesting study; however, certain improvements are necessary to enhance its quality. Below are my recommendations:
-
The abstract is highly confusing and should be revised for clarity.
-
Sections 4.2 and 4.4 lack essential details regarding the critical parameters used in the experiments to study the effects of fatty acids in both 2D cell monolayers and in vivo models. I suggest the authors provide additional information, such as:
- The final concentration of fatty acids used in both cell cultures and mice.
- Whether the same concentration was applied across all cell lines.
- The incubation time for the cells.
- Did the authors evalaute the cell viability after the incubation period?
- Whether the authors assessed the inflammatory effects of the fatty acid mix in cells, similar to the initial in vivo experiments, or if they only analyzed the effects of individual fatty acids, as reported in Figure 5. Additionally, what was the rationale behind this approach?
- Whether the concentrations of fatty acids used in mice were the same as those applied in cell cultures.
-
Another critical aspect requiring clarification is the method used to measure the gray coloration of the mice shown in Figures 3e and 4b. This information is not provided in the Materials and Methods section and should be included.
Round 2
Reviewer 2 Report
Comments and Suggestions for Authors
The revised version is now ok.